# Misincorporation Proteomics Technologies: A Review

**DOI:** 10.3390/proteomes9010002

**Published:** 2021-01-21

**Authors:** Joel R. Steele, Carly J. Italiano, Connor R. Phillips, Jake P. Violi, Lisa Pu, Kenneth J. Rodgers, Matthew P. Padula

**Affiliations:** 1Proteomics Core Facility and School of Life Sciences, The University of Technology Sydney, Ultimo, NSW 2007, Australia; Joel.Steele@uts.edu.au (J.R.S.); jake.violi@uts.edu.au (J.P.V.); 2Neurotoxin Research Group, School of Life Sciences, The University of Technology Sydney, Ultimo, NSW 2007, Australia; carly.italiano@uts.edu.au (C.J.I.); connor.phillips@student.uts.edu.au (C.R.P.); Lisa.Pu@student.uts.edu.au (L.P.); kenneth.rodgers@uts.edu.au (K.J.R.)

**Keywords:** misincorporation, non protein amino acids, post translational modifications

## Abstract

Proteinopathies are diseases caused by factors that affect proteoform conformation. As such, a prevalent hypothesis is that the misincorporation of noncanonical amino acids into a proteoform results in detrimental structures. However, this hypothesis is missing proteomic evidence, specifically the detection of a noncanonical amino acid in a peptide sequence. This review aims to outline the current state of technology that can be used to investigate mistranslations and misincorporations whilst framing the pursuit as Misincorporation Proteomics (MiP). The current availability of technologies explored herein is mass spectrometry, sample enrichment/preparation, data analysis techniques, and the hyphenation of approaches. While many of these technologies show potential, our review reveals a need for further development and refinement of approaches is still required.

## 1. Introduction

The “central dogma” of molecular biology suggests that the translation of one gene results in the expression of a single protein [1]. However, translated proteins are known to exist as multiple biological variants or proteoforms [2]. These proteoforms are the result of modifications to the polypeptide chain, including the addition, subtraction, or alteration of chemical groups. Such modifications can endow proteoforms with biological activity or an altered function varying from the original proteoform [3]. Any modification that occurs to a proteoform once already translated is termed a post-translational modification (PTM), generating a new and distinct proteoform, adding the advantage of complexity to the proteome [4,5,6].

Additionally, and importantly for this review, variant proteoforms may be generated by mechanisms other than post-translational modification. During protein translation, an incorrect amino acid may be inserted into the growing peptide chain, resulting in a modification in the final proteoform. Such errors generate new, non-native proteoforms that have the potential to cause harm to the cell [7,8]. The mistranslational error rate reported in *Escherichia coli* (*E. coli*) is between 0.5–5% at any amino acid position [8]. This occurrence of errors in translation during synthesis can generate a new proteoform in the same way that cleavage or PTM can e.g., N-terminal methionine excision [8,9,10]. However, instead of providing an advantage to the cell, proteoforms produced through mistranslation often present a burden, as they are unpredictably generated with the resulting non-native proteoforms prone to misfolding [11,12]. To overcome this, the cell has sophisticated machinery to identify and degrade these proteoforms [13].

The “misincorporation” of incorrect amino acids into a proteoform need not be limited to the 20 canonical amino acids used in proteoform synthesis. Thousands of synthetic and naturally occurring nonprotein amino acids (NPAAs) exist, also referred to as “nonproteogenic”, “noncanonical”, “noncoded”, or “non-natural” [14]. The infiltration of NPAAs into the protein translation process adds further opportunity for mistakes to be made during translation, resulting in distinct and unintentionally produced proteoforms. Such misincorporation can cause aberrant modifications to proteoform structure, with biological ramifications for the cell or organism that can perturb normal cellular function. It has been shown that misincorporation of an NPAA can alter the 3D structure of a proteoform, resulting in aggregation [15]. Such misfolding and aggregation of proteoforms is known to be a hallmark of numerous degenerative neurological diseases [16] and for some diseases, an association already exists between exposure to NPAAs and disease development. This includes motor neuron disease (MND) [17,18], multiple sclerosis [19,20] and neurolathyrism [21,22]. As such, the exploration of NPAA misincorporation into proteoforms to date has largely focused on their underlying potential to trigger proteinopathies and cause neurodegeneration. To describe the study of NPAA misincorporation into the proteome, we introduce the term Misincorporation Proteomics (MiP), whereby during protein translation, a genetically encoded canonical amino acid is replaced by a NPAA.

## 2. Amino Acid Misincorporation

Protein translation involves the cognate amino acid (AA) being charged by its appropriate aminoacyl tRNA synthetase (aaRS; Figure 1). The charging is enabled by the hydrolysis of adenosine triphosphate (ATP) and allows the formation of an AA/aaRS/AMP complex. The cognate transfer RNA (tRNA) then binds to this aaRS/AA/AMP complex and an aminoacyl ester bond forms, transferring the aminoacyl group to the tRNA and releasing AMP [23,24].

Correct binding of amino acid to its aaRS is dependent on the fit between the amino acid and enzyme binding pocket [25]. If an incorrect amino acid binds the aaRS, this can be removed by a series of proofreading functions known as pre- and post-transfer editing (Figure 2). Pre-transfer editing occurs before tRNA binds to the charged amino acid. Removal of the charged amino acid has been suggested to occur via various mechanisms, including selective release by the aaRS enzyme, translocation to a separate hydrolytic editing site on the enzyme, or hydrolysis at the primary active site of the aaRS [25,26]. Post-transfer editing occurs after the mischarged amino acid is attached to tRNA and involves the hydrolysis of the ester bond between the two. This can occur via cis-editing at a separate editing domain, by trans-editing factors that resample mischarged amino acid tRNA at the aaRS, or by free trans-editing factors [25,26]. The extent to which pre- and post-transfer editing occurs in human aaRS enzymes is poorly understood, as are the mechanisms by which pre- and post-transfer editing select for misincorporations of NPAAs in particular.

Despite the presence of checks to avoid incorrect amino acid incorporation, misincorporation can still occur. The misincorporation rate at any amino acid position is quoted as 1/10,000 when considering only the 20 canonical amino acids [9]. However, this rate may be higher when considering NPAAs could be present within the available pool of amino acids. The misincorporation of an NPAA into a growing polypeptide chain through its mischarging onto the aaRS, followed by the failure of the aaRS to remove the mischarged NPAA, allows for the non-native proteoform to be released. Additionally, an NPAA may outcompete the canonical amino acid for binding, especially when there is a higher concentration of NPAA present [15,18,27,28]. Depending on the position of NPAA misincorporation and the canonical amino acid that was replaced, there may be a conformational change in the proteoform, which predisposes it to misfolding and aggregation. Other issues may include replacement of amino acid residues that are essential sites for activity or post-translational modifications (PTM) in that proteoform. Such alterations to the amino acid sequence can hinder a cells ability to operate, as the proteoforms may be nonfunctional, aggregate, or even acquire a toxic gain of function. An example of the impacts which NPAA misincorporation may cause is the neurological disease multiple sclerosis, the development of which has hypothesised links to misincorporation of the NPAA azetidine-2-carboxylic acid (azetidine), a proline analogue derived from the common agricultural crop sugar beet (beta vulgaris), into myelin basic protein [27]. The consequences posed by the misincorporation of NPAAs may be broad and unpredictable, making it important for the cell to have defences and solutions to this problem.

In addition to the checks during polypeptide synthesis that help to both initially avoid the mischarging of amino acids and to hydrolyse any mischarged amino acids from a tRNA, there are also checks in place following polypeptide synthesis that detect and remove misfolded proteoforms. Chaperones attempt to refold proteoforms to the correct conformational structure and if unsuccessful, degradation pathways such as the proteasome and autophagy are utilised [28]. The failure or bypassing of these checks altogether should trigger cell death. The toxic species of proteoform produced by the misincorporation process may aggregate and self-propagate, forming the basis for proteinopathies and NPAA pathologies [29,30,31]. As the induction of apoptosis/necrosis occurs when the burden of protein aggregates reaches a threshold, this presents a bigger problem for terminally differentiated cells such as neurons. Such cells cannot share the burden of aggregated proteins amongst daughter cells by dividing and must bear the burden fully, increasing their susceptibility to NPAA pathologies [31]. This may explain why diseases associated with NPAAs tend to be neurological in nature, and why it is of growing importance to investigate misincorporation and its consequences.

## 3. The Identification of NPAAs Misincorporation

The measurements of NPAA misincorporation to date have primarily relied on inference methodologies, indicating NPAA presence but not location, rather than the direct sequence localisation of a peptide or protein containing a NPAA (Table 1). Studies by Fowden and Richmond [32] in the 1960s were part of the foundational works for the exploration of NPAA misincorporation [33,34,35]. Here, NPAAs are recognised as structural analogues of canonical amino acids [36] with various toxic effects and as metabolites arising from their mimicry and competition with the canonical amino acids they resemble. This resulted in the term “antimetabolite” being used to describe these toxic NPAAs, with the arginine analogue Canavanine exemplifying this [37].

Various methods were employed during these early studies to investigate the misincorporation of NPAAs. This included the addition of NPAAs and canonical amino acids in different ratios to growing bacterial cultures to observe toxicity and infer competition between NPAA and canonical amino acids [32,38,39]. Such studies offered insight into the similarity of some NPAAs to specific canonical amino acids and therefore an understanding of what pathways would be most affected by the NPAA. Since these early studies, there has been a range of methods used to infer mistranslation and misincorporation of amino acids generally, including NPAAs. Some approaches included studying misincorporation at the tRNA level using radiolabelled amino acids and tRNA microarrays to detect misacylation [40]. Such methods have shown that the NPAA Beta-methyl-amino-L-alanine (BMAA) charges to both alanine and serine tRNAs and bypasses the proof-reading ability of the alanine aaRs, suggesting misincorporation at alanine positions [41,42]. An alternate approach that has proved popular is the use of radiolabelled amino acids, particularly in *E. coli*. Detection of a radioactive signal within isolated protein infers that the labelled amino acid has been incorporated, replacing the canonical amino acid, or is associated with the protein fraction. Examples include detection of radiolabelled cysteine in cysteine-free flagellin in *E. coli* [43]. Similar studies in *E. coli* [44,45] and rabbit reticulocytes [46] have also used this method.

Another method of inferring amino acid misincorporation is to study the effects on the proteoforms themselves if misincorporation occurs. This includes measuring changes in the isoelectric point of a proteoform due to different amino acid residues being present [47], or restoration of enzymatic activity, including fluorescence, of the resulting proteoform species [48,49,50,51,52,53]. Other methods include inference of incorporation by an inability to detect free COOH or NH2 groups of the NPAA unless first hydrolysed from protein (dinitrophenyl assay). Alternatively, the use of detectors coupled to chromatographic and ionophoric separation have also been used for the identification of NPAAs from a protein hydrolysate [36]. While these various techniques have provided useful information, many also rely on the use of bacterial systems, limiting their application in mammalian research. Most importantly, all these methods offer only an indirect measurement of misincorporation and cannot definitively characterise the misincorporations of NPAAs. For a review of these techniques, refer to the work of Ribas de Pouplana et al. [54].

There are also direct methods of detecting misincorporation that involve mass spectrometry. This provides the opportunity to localise a NPAA in a peptide sequence and identify misincorporation of incorrect amino acids based on side-chain modifications or the use of a modified database algorithm [55,56]. Identification of NPAAs in the hydrolysate of protein fractions via high-pressure liquid chromatography (HPLC) coupled to either mass spectrometric (MS) (including tandem (MS/MS)) [41], or spectrophotometric detectors (such as ultraviolet-visible spectroscopy (UV-VIS)), is a routine technique. Additionally, there has been an analytical method developed for assaying misincorporation in overexpressed proteins in *E. coli* and yeast called MS-READ, which utilises a genetically modified overexpression model combined with affinity purification [25]. In a similar manner, ESI-MS has been used to study the incorporation of L-3,4-dihydroxyphenylalanine (L-DOPA) in proteins expressed in *E. coli*. However, although a mass shift corresponding to L-DOPA was observed in the protein and tryptic peptides, fragmentation of the trypsin-generated peptides was not performed, preventing localisation of the L-DOPA in the peptide sequence [57]. The use of hydrolysis and detection methods indicating NPAA presence do not provide proteoform sequence information which is essential in studying the effects of NPAAs on biology.

The sensitivity of the methods that can be employed to explore misincorporation varies with the more sensitive methods sacrificing direct site localisation for a decreased limit of detection. These highly sensitive methods include radiolabelling experiments, amino acid analysis of protein hydrolysate, ELISAs and antibody-based microscopy. These methods can be employed alongside exploratory proteomic methods and the usage of targeted mass spectrometry for quantification can also be employed following identification of misincorporation species. When applied to the quantification of the misincorporation species, indirect methods like amino acid hydrolysis have sensitivity in the parts-per-billion (ppb) or pictogram range. Antibody-based detection methods using coupled enzyme-based reporters, such as horseradish peroxidase (HRP), could theoretically detect a single antigen molecule if enough substrate is converted to a detectable product but detection limits will vary with antibodies and variation of molecular structure around the NPAA site.

As the advancement of proteomics has led to an increase in the ability to detect and characterise canonical amino acid mistranslation, the applications of the techniques have diversified. This review will firstly outline key considerations in the exploratory analysis of MiP, including experimental considerations and biological assumptions that should be understood before utilising certain methods and models. Secondly, a summary of the mass spectrometric technologies that have become available in the pursuit of investigating these misincorporations will be discussed, including targeted global proteomics, data-independent acquisition technologies, precursor ion scanning, and unique instrument system configurations. Considerations for sample processing and potential enrichment strategies that can be employed or require development are introduced before a discussion of the diverse range of pipelines, processes, and data analysis suites originally designed for PTM identification that can be repurposed for MiP. While the bulk of this paper will focus on the ramifications for human samples, many of the techniques discussed are still applicable to a large array of different sample types. From the outset of a study, it is essential to understand that for the generation of high-quality data in MiP, there needs to be enough of the NPAA misincorporation in a proteoform at a specific location prior to protein extraction as analysis methods in proteomics, such as gel electrophoresis and mass spectrometry, are concentration sensitive technologies.

## 4. Key Considerations in Mistranslation Proteomics

The nature of NPAA misincorporation is random or stochastic and when considering a single site of incorporation, and excluding mechanistic interference, the proportion of proteoforms containing a misincorporation will be several fold less abundant given the highest observed error rates of 10% in canonical amino acid substitution at a single site substitution (Asn->Asp) in *E. coli* [55]. Additionally, detection becomes even more difficult when considering that misincorporation may randomly occur at more than one site in a peptide sequence, resulting in a wider variety of proteoforms, and thus peptidoforms, all at a lower abundance. For example, an incorporation rate of 1 in 1000 with three possible sites would have a 1 × 10^−12^ abundance in comparison to the unmodified molecule, which would be beyond the limits of detection for the most advanced mass spectrometers. This makes the detection and subsequent investigation of the mistranslated species unlikely without utilising specific extraction and pre-concentration/sample enrichment techniques.

Another important consideration for the study of misincorporation is the suitability of the models (yeast/*E. coli*/mammalian cell lines) and the time frame which is appropriate for sufficient incorporation for detection versus the transient or acute effects on the proteome. The use of animal models could be employed but resource limitations and time constraints are often beyond that of most laboratories. While this review is focused on the study of misincorporation, the investigation into misincorporational effects on the proteomic landscape would also be important, as this could determine the underlying pathology necessary to classify systemic disease pathways (NPAA-pathology). This may help elaborate biological cases arising from exposure to NPAAs and help build upon biomarker classification for individual NPAAs. When considering the complexity of NPAA pathology, studies in cellular models may be limited in their application to human disease processes, especially when perturbation of PTMs that have biological function occur Table 2.

Detection of NPAA incorporation in humans may only be possible in individuals exposed to a subtoxic intake of a NPAA over a prolonged time of many years i.e., supplementary intake of norvaline in bodybuilders [59], azetidine (from beta vulgaris) in meat and milk consumers [60], and Parkinson’s disease patients undergoing L-DOPA therapeutic treatment [15]. However, when compared to cell lines and bacterial models, samples are not as readily available. Choosing overexpression models is not a biologically relevant methodology but it will give an approximate representation of the incorporation rate of NPAAs, and researchers should use caution when drawing inferences between organisms and human disease. The impact on cellular systems in multicellular organisms will be different from that of single cellular models and will not consider the larger repertoire of proteostasis mechanisms.

## 5. Sample Processing and Enrichment

Sample processing and enrichment techniques are critical to ensuring that downstream analytical methods have the optimal chance of detecting low abundance proteoforms, such as those containing NPAAs. All the techniques presented here can increase the chance of identification, however, the success of all MiP experimental approaches will be limited by the amount of the NPAA containing species in the sample. Therefore, most explorations of NPAA’s have relied upon inference methods showing their presence but not sequence location via methods including radiolabelling and amino acid assaying of hydrolysed proteins, rather than direct localisation [40,41,42,43,44,45,46,48,49,50,51,52,53,54,55,57]. The use of fractionation, enrichment, and depletion are essential, but the caveat is always the potential loss of target proteoform due to the sample processing. As an example, the starting amount of protein used to examine the acetylated proteome was reported as 15mg to assay several thousand peptides [61], whilst typical shotgun proteomic experiments are performed on several micrograms. To this end, quantifying the amount of NPAA using radiolabels or hydrolysate amino acid analysis can infer the amount of starting protein that is required and if the NPAA is indeed present in a sample prior to the MiP workflow employed.

The enrichment of proteoforms containing a NPAA can employ various strategies including; antibodies (pull-downs), metal and chemical affinities (chromatography), or fractionation that concentrates targeted species. Antibody-based affinity enrichment is often the first choice of enrichment for PTM studies [62] and requires a highly specific antibody. Antibodies are time-consuming and expensive to create, especially for small discrete chemical groups such as an amino acid side chain [63,64], therefore manufacture requires considerable supporting evidence of specificity. Enrichment strategies such as metal affinity as used in phosphopeptide enrichment may enrich an NPAA containing species, although its application will be limited to those NPAAs whose chemistry is amenable to binding.

Fractionation methods can help detect low abundance proteoforms or peptides and the methodologies generally applied are based on molecular weight, charge, and hydrophobicity. The aforementioned methods do not specifically target peptides containing NPAAs but may give a better chance at identification by dynamic range reduction and increase in the amount of a specific proteoform or peptidoform. As NPAA incorporation may be implicated in proteinopathies and neurodegeneration, enrichment of NPAA containing proteoforms by isolation of insoluble proteoforms from cell lysates is an attractive selection method that has been employed for such protein aggregates as β-amyloid and Tau [65].

Approaches to create sufficient levels of NPAA-containing proteins for detection include the use of in vitro systems whereby protein production can be controlled, and high levels of NPAAs introduced. The limitation with such approaches is that the organism level disease pathology is not seen. An example of these methods is MS-READ [25] to study NPAA incorporation into over-expressed GFP-elastin protein in *E. coli* or yeast. This methodology is quoted as providing quantitation down to the 1/10,000 mistranslational rate for canonical and NPAA species, applying the use of affinity columns to the GFP-elastin protein and then chromatographic separation of all the captured species. This model can establish a baseline of NPAA incorporation, but a mammalian cell line used in the same manner will provide a closer estimate of the rate of incorporation specific to human tRNA binding/proofreading. The NPAA BMAA has been investigated by this method alongside tRNA binding studies that implicated incorporation in place of alanine (contrary to past research identifying the site as serine [29]), with the product being un-quantifiable [42]. Further complications arise when model systems such as *E. coli* or yeast have alternate biological responses to an NPAA compared to mammalian systems. For example, the response and misincorporation dynamics of BMAA in *E. coli* appears markedly different than in mammalian cells [41], and *E. coli* are reported to have yet unidentified mechanisms of avoiding misincorporation of this NPAA [66]. Quantitative analysis of NPAA-modified versus native proteoforms is not currently possible as the ionisation and chromatographic profiles (retention times) between the two peptide or proteoform species will differ. The amount of the modified proteoform can be inferred from a loss of intensity in the unmodified species relative to the abundance of the other proteotypic peptides assigned to the open reading frame, as shown in Figure 3. The synthetic modification of all species is possible in certain instances where the chemistry is simple, such as the addition of a hydroxyl group to Tyrosine to form the NPAA L-DOPA with the subsequent use of heavy labelled oxygen supplied via peroxide to allow relative comparison. This method has been applied for methionine oxidation [67]. In the same manner, the synthesis of NPAA-containing peptides could be used to quantify the amount if the target species is known. This approach has been applied to the study of the entire human proteome by the Kuster lab and JPT peptides, who have synthesised all human peptide sequences and acquired LC-MS/MS orbitrap data to form a pan-human spectral library and the tool PROSIT [68].

Abundance is only directly comparable between identical species. Conversely, the ionisation efficiency of the modified sequence will not reflect the missing amount of unmodified peptide. Thus, only a decline in the abundance of the unmodified peptide can be quantified accurately rather than the increased abundance of the NPAA-containing peptide.

The reader is cautioned against the use of peptide labelling procedures used for relative quantification by multiplexing samples, i.e., Tandem Mass Tags (TMT) [69] and Isobaric tags for relative and absolute quantitation (iTRAQ), in the identification of NPAA containing species [70]. The NPAA containing peptides do not exist in a control sample and by combining multiple proteomes the signal of these unique species will be diluted. This may be why Beri et al. [71] were unable to identify BMAA-containing peptides when searched as a serine modification, however recent data suggests that the modification may be in place of alanine [72].

It is important to note that if starting material is scarce, that sample processing techniques requiring higher starting material should not be used. As such, the development of specific sample processing techniques for studying MiP is equally as important as the mass spectrometric technologies employed.

## 6. Mass Spectrometric Technologies

The current state of the art in proteomic technologies has lowered the amount of sample required to less than a microgram for a comprehensive shotgun proteomics analysis, however the linear dynamic range has stretched marginally to approximately four-five orders of magnitude in orbitrap mass spectrometers [72]. There are many ways to increase sensitivity to identify and selectively sequence these lower abundance ions. For confident spectral identifications and resolving of parent mass ions, a High-Resolution instrument like that of an FT-ICR/orbitrap or high-resolution Quadrupole-Time of Flight (Q-TOF) should be used.

Generally, the peptides containing NPAAs are orders of magnitude lower in abundance in comparison to the total proteome. As such without sample enrichment strategies, novel instrument operation techniques, correct ion dissociation techniques, and the hardware being able to intelligently operate, identification is simply not feasible. Herein, the technologies concerning the mass spectrometer that can increase the identification of the low abundance misincorporations will be covered: Mass spectrometer base requirements; Untargeted (either Data-Dependent Analysis (DDA) or Data Independent Acquisition (DIA)), Targeted single reaction monitoring (SRM) and multiple or parallel reaction monitoring (MRM/PRM), hybrid methods, and dynamic range optimisations.

## 7. Mass Spectrometer Base Requirements and Desirable Features

The advent of high-resolution accurate mass (HRAM) instruments has enabled the discrimination of low abundance ions from background noise especially in the instances of FT-ICR/FT-orbitrap. The higher resolution settings on the orbitrap of 240,000–1,000,000 FWHM can determine the exact elemental composition of an ion. When applying this to peptide sequences, it becomes feasible to accurately identify not only the sequence but also other chemical properties of the peptides. For example, the characteristic fragmentation patterns under various energies and fragmentation types, revealing the presence of PTMs.

In Orbitrap instruments, this high resolution is usually sacrificed in MS2 acquisition for speed as 17,500 resolution is used in standard DDA methods, in conjunction with the high-resolution MS1 scan, allowing an accurate assignment of fragments to amino acid sequence. Within the isolation window for precursor transmission accumulation and fragmentation (1.4 Da in an orbitrap (± 0.7 Da)) there is a potential that the fragmented species may be derived from more than one single ion population given the likelihood of similar *m*/*z* ion species being found within the isolation window selected for fragmentation. This produces chimeric fragmentation spectra that may confuse the correct assignment of the precursor’s structure, although modern peak-picking algorithms somewhat overcome this. This is critical when an investigation of MiP requires accurate assignment. A way to further overcome this issue is BOXCAR which is explored below [73].

## 8. Data Dependent Analysis

Current untargeted exploration of the proteome by LC-MS/MS involves the use of Data-Dependent Acquisition (DDA), a method of operating the mass spectrometer to sequentially select from an initial precursor scan of the most abundant precursor species for fragmentation. This mode of operation is an important foundation for data generation as each MS/MS spectrum should be derived from a single precursor (excluding the aforementioned chimeric spectra). However, DDA favours sequencing of the high abundance peptides in the sample, precluding the detection of low abundance species without prior offline enrichment, depletion, or fractionation. However, there are several ways to increase the depth of observed ions by the addition of inclusion/exclusion lists to detect lower abundance ions, but the generation of these lists requires prior knowledge of the sample and NPAA containing species.

Due to the nature of ion accumulation and space charging effects, only a certain number of ions can be measured in the MS1 scan and lower abundance ions are masked by the presence of higher abundance ions [74,75]. The way in which one can identify low abundance species, particularly of a NPAA containing peptide, is to limit the precursor mass range using gas-phase fractionation to break up the ion current into smaller ranges [76]. This would turn a traditional acquisition using a 300–1500 *m*/*z* range into 2–7 mass ranges, which are subsequently combined bioinformatically.

A more robust approach to sampling the ion current and limiting the dynamic range transmitted is the usage of BOXCAR. Developed by the Mann lab, this can increase the dynamic range ten-fold for the detection of ions [77]. This methodology uses a precursor ion scan to survey the ion species distribution across the mass range, with the mass range subsequently divided into windows with each having the same ion auto gain control (AGC) target. The number of these BOXCAR scans can be increased to further increase the number of lower abundance ions that can be accumulated by narrowing the window while keeping the same AGC target (which in our experience is three). This method is excellent for finding low abundant species but does not increase the ion accumulation of low abundance species for a MS/MS scan, increasing the probability of their identification. Furthermore, the BOXCAR scans occupy a significant amount of duty cycle time, limiting the number of precursor ions that can be fragmented. Jenkins and Orsburn’s preprint on BOXCAR-assisted mass fragmentation (BAMF) offers key insight into how partial ion stream sampling can increase the depth of identification when combined with fragmentation [75]. The benefit of low abundance ion identification is that it allows larger spectral libraries to be created and queried. A realised version of this methodology was also developed by the Gygi lab and similarly limits transmission by using parallel-notched waveform isolation [78].

The incorporation of targeting can be used concurrently in a DDA experiment and added to the mass spectrometer’s cycle time. The usage of targeting lists requires prior knowledge of the peptide species that could contain an NPAA, and this can be generated in silico by taking currently identified species of highly abundant and confidently matched peptides, then adding the mass difference that incorporation would theoretically produce to create a theoretical precursor mass for targeting. However, without knowledge of the chromatographic elution profile, the target list cannot be scheduled, and thus the length of the list must be restricted to manage duty cycle with expected chromatographic peak width to ensure sufficient measurements across the peak, as well as differentiating isobaric peptides being detected at differing retention times over the duration of chromatographic separation. This issue of ions similar to the targeted precursors being detected during the chromatographic separation can be overcome by using the targeting mode within MaxQuant Live [79] that can dynamically warp global targeting lists. Dynamic warping of targeting lists combats retention time drift that is apparent in intra sample injections as the gradient conditions may have been slightly altered causing drift and a potential loss of the targets during a separation. There is also a function built-in called BatMode enabling fragmentation of the targeted precursor mass during every duty cycle for the predicted elution window. This method utilises mass inclusion lists for fragmentation which can be implemented on any Q-Exactive mass spectrometer that can perform DDA and handle targeting schemes, which in turn decreases the limits of detection.

## 9. Data Independent Acquisition

Data independent proteomic approaches offer an increased depth of quantitative analysis with the ability to generate a permanent digital ion record of the entire sample being analysed [80]. The drawbacks of this technology are the reliance on generated spectral libraries from DDA sample analysis, which are necessary for searching, and the computational overhead required to generate them. For human samples, a synthetic library of peptides that have been fragmented in an orbitrap instrument has been used to create the PROSIT tool [68]. This allows the researcher to download a spectral library based on a FASTA file provided and align it to the retention times of the instrument. This increases the identification rate by using “match between runs” in various software and increases the richness of the data acquired as exemplified by studies on the PTM phosphorylation [81].

Unfortunately, NPAA containing peptides are not present in the various databases of PROSIT and SRM/human peptide atlas. Acquiring data in a DIA manner will allow retrospective interrogation for these peptides, with a caveat that the amount of NPAA-containing peptides present for analysis must be above the lower limits of detection. The ways to overcome the lack of theoretical data available to predict the effect of NPAA incorporations on produced MS/MS spectra is through the use of tools (such as DeepLC [82]) that enable the prediction of retention times and elution windows for these unknown species. This allows smarter scheduling and the ability to potentially target tens of thousands of precursors in a single injection. In the data analyses section of this manuscript, the tools that can be used for in silico mining will be covered.

## 10. Immonium Ion and Precursor Ion Scanning

The use of precursor ion scanning in combination with stable isotope labelled amino acids was developed by Purcell and Williamson [83]. This technique relies upon a diagnostic immonium fragment ion that is unique to an amino acid. In the classical method, stable isotope labelled analogues are used in cellular treatment and produce mass shifted product ions. In this instance, it relies upon the NPAA to produce a unique product ion such as an immonium ion (a fragment resulting from the amino acid side chain). If an immonium ion is formed from the NPAA of interest, it is then feasible to use a stable isotope containing NPAA, such as N-15, during treatment to produce this “diagnostic ion”. The size of certain immonium ions may be below the scanning range in most general shotgun proteomic experiments, which may require specific operational parameters and precludes the use of certain instruments.

## 11. Ion Mobility Mass Spectrometry

A highly desirable feature for the analysis of MiPs and even PTMs is ion mobility separation devices (IMS) available on several state-of-the-art mass spectrometric platforms, with most major vendors offering a variation. For a review on the different types of IMS please see: [84]. For further reading, see [85,86,87] and for example of novel usage, refer to [88]. In theory, NPAA containing sequence isomers could have the same *m*/*z* and potentially same elution time in chromatography producing chimeric fragmentation spectra and being indistinguishable in traditional proteomic LC-MS/MS methods. IMS offers the ability to further separate ions using the third dimension of Collisional Cross Section (CCS) to separate isobaric, co-eluting peptides. While there are several variants of IMS, there are three variants commonly used with MS/MS that would be best suited: traveling wave ion mobility spectrometry (TWIMS), drift-time ion mobility spectrometry (DTIMS), and trapped ion mobility spectrometry (TIMS). TWIMS and DTMS determine mobility via how the ions drift in a cell filled with gas, with the only difference being the application of voltage in DTMS. TIMS determines mobility by trapping the ions in place and having gas pass through the cell. IMS may overcome chimeric spectra production and distinguishing precursors and products [74] but its immediate usage is to increase the depth of analysis of a sample by fractionating the ion current. Implementing any of the approaches mentioned within this manuscript in combination with IMS could provide high-quality empirical data for the incorporation of NPAAs.

## 12. When Is an Incorporation Real?

When performing PTM or MiP analysis, it is important to establish parameters for correct positive identification of a modified spectra that are sensible. The requirements are a statistically significant peptide sequence identification, based on robust and established statistical analysis and false discovery rate determination [11,89,90,91,92,93,94]. Positional fragments for the NPAA in an MS/MS spectrum are mandatory for site localisation, similar to the Ascore developed to account for sequence isomers in phosphoproteomic experiments [95]. On the rare occasion that sequence isomers elute together and form a chimeric spectrum, it may not be possible to assign a localisation for the site of misincorporation. The next level of identification should be that of a spectrum matching the unmodified peptide existing within the sample, showing that the peptide belongs to the parent protein or open reading frame. The prime consideration should be the peptide’s presence exclusively in the conditions where misincorporation can occur otherwise, an identification is likely a false positive. As the search space is increased to allow multiple variable modifications on a single peptide, the chances of a forced match/false positive occurring become statistically likely.

## 13. Data Analysis Techniques

Once the acquisition of data has been performed, the most time-intensive part of a proteomic experiment begins, that of data analysis. A typical proteomics experiment will employ packages or pipelines that will perform the general required processing, precursor/fragment extraction, centroiding, and mass recalibration. The spectra will then either be searched directly against an in-silico generated tryptic digest database created from the FASTA files for the relevant organism/s and Peptide-Spectrum Matches (PSMs) validated through statistical models or through de novo sequencing of the spectra performed with or without reference to a database.

The repurposing of proteomic tools to specifically explore MiP involves the mass shift of the NPAA incorporation being known at each position of incorporation. Tools that can consider the NPAA as a “PTM” by indicating it as a variable modification in traditional database searching is possible if incorporation is at a high enough abundance to produce an MS/MS spectrum of sufficient quality and the substitution position is correctly characterised. These mass shifts for literarily relevant NPAA incorporations are listed in Table 3, and for further reading on NPAAs, see [96,97,98]. Traditional database searching is heavily reliant on the database using a known variable modification to identify NPAA species and as such, a method known as “open” searching is recommended for MiP exploration as it allows an exploration of the NPAA’s presence at multiple sites and positions in an unbiased manner. The underlying flaw of traditional database searching is still not resolved using spectral libraries in DIA. For identification of a NPAA containing species to be possible in DIA the reference database still must have identified the peptide, it is foreseeable that bioinformatically this problem could be addressed by the assignment of unassigned, co-eluting set of fragments that perfectly align in retention time in relation to the identified unmodified species.

Several programs that have been developed that can perform open searching on datasets. These programs have developed ways to restrict the computational search space considered to localise PTMs in unmatched spectra. The main examples include; Fragpipe [109], MetaMorpheus [91], Open-pfind [110], tagGraph [111], Peaks Studio [112], Proteome Discoverer [113], Byonic [114], MaxQuant [92], for more examples see Table 4. These programs are essential to the exploration of NPAA localisation within the proteome. As an example, we will cover a method that has been developed for systematic detection of amino acid substitutions that does not rely on the use of a genetically altered or modified cellular expression system [8,25,42] and does not require the use of stable isotope labelling [115].

The methodology developed by Mordret et al. [11] relies upon a “blind modification” search and repurposes the MaxQuant “dependent peptides” search to explore amino acid substitutions [129]. This methodology could be repurposed in a NPAA exposure model with subsequent identification of incorporation, given a large enough starting sample with known NPAA protein association. This methodology employed two forms of fractionation, solubility and strong cation exchange, which can be routinely performed by most laboratories. The peptide search relies on the principle that the modified peptide sequence will be of lower abundance compared to the identified unmodified counterpart, the peptides that are assigned to known modifications from the Unimod database are filtered out and peptides remaining unexplained are considered for NPAA incorporation. Within Mordret’s paper they were also able to align chromatographic elution profiles for substituted amino acids which could be used to determine the elution profiles of NPAA species. This paper provides the best pipeline for analysing NPAA containing proteomes and is highly recommended for the investigation of the MiPome.

## 14. Conclusions, a Future Direction and a Best Practice for MiP

Outlined throughout this paper are ways to increase the identification of NPAA containing species and have framed this as mistranslational proteomics or MiP. Key consideration should be given to the sample selection and model creation that offer the highest level of misincorporation, and it is advised that the investigating lab employs amino acid analysis on the proteome prior to committing resources to proteome analysis. The next step in a workflow is to employ an enrichment method or fractionation to further reduce the dynamic range of protein abundance. It is noted here that development of a specific enrichment method may be required, either chemical or an antibody-affinity based. Investigators are also cautioned against the use of peptide labelling tags for use in NPAA peptide identification.

It was outlined that operation of a mass spectrometer that can transmit a smaller mass range of ions to increase the dynamic range, such as that of BOXCAR or parallel-notched wave form isolation, will enhance the depth of identification and that IMS can be used to further separate out NPAA containing species from higher abundance peptides and separate chimera producing sequence isomers. The data analysis workflow is the key to whether a NPAA containing species will be identified, and several programs are listed that can perform open searches that can be used to explore the MiPome. Furthermore, a complete pipeline was identified that can be employed for the investigation of NPAA proteoforms systematically without the requirement of genetically modified models or use of expensive stable isotopic reagents.

The further progression of the entire field of proteomics towards the study of proteoforms will continue to increase the ability to detect and quantify the MiPome, which will provide insight into how disease related to NPAA incorporation progresses and infer potential treatments.

## Figures and Tables

**Figure 1 proteomes-09-00002-f001:**
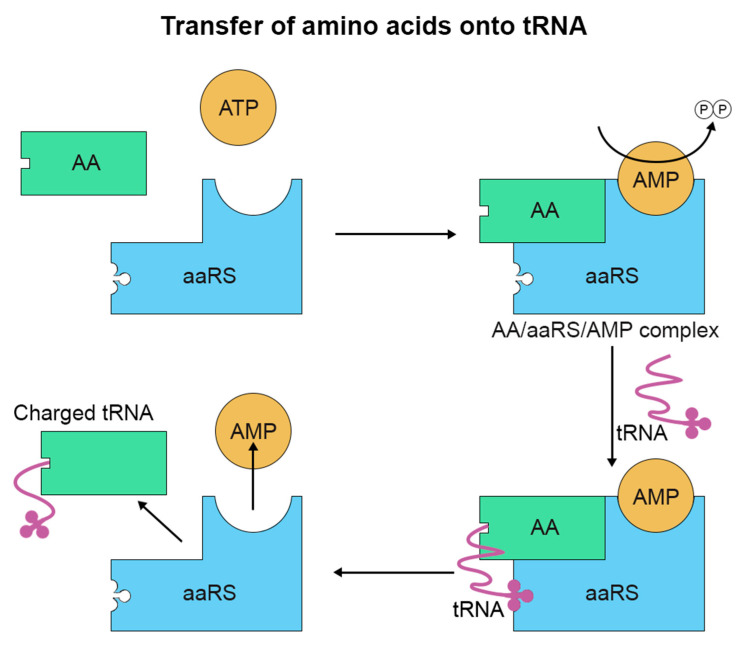
The transfer of amino acids (AA) onto tRNA through tRNA synthetase (aaRS) in the presence of ATP. ATP is converted to AMP activating the aaRS with the AA, an tRNA is then bound to the AA of the AA/aaRS/AMP complex subsequently the now charged tRNA dissociates.

**Figure 2 proteomes-09-00002-f002:**
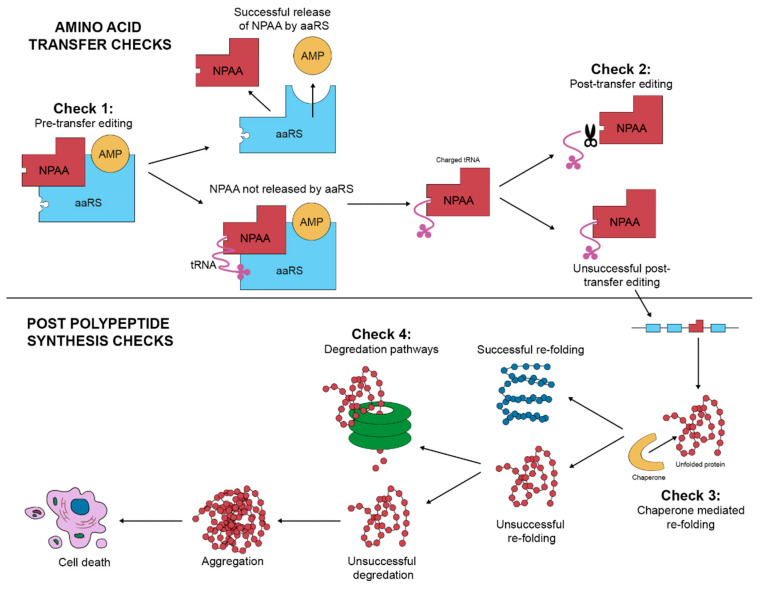
Checks of mistranslation within a cell. AAs are checked for correct charging pre- and/or post-transfer. Misfolded species can be refolded via chaperones or degraded by the various proteolytic pathways; aggregates that are not degraded can lead to cell death.

**Figure 3 proteomes-09-00002-f003:**
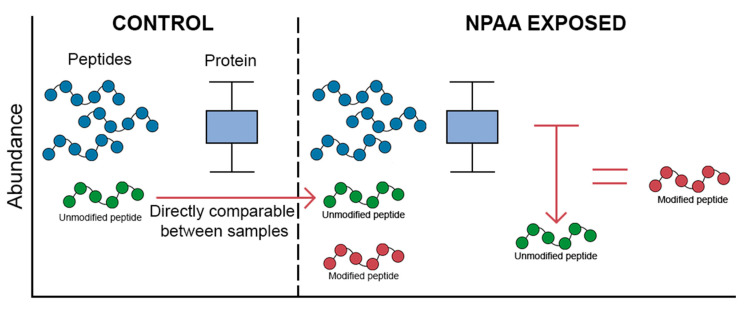
How to compare mistranslated sequence abundances. Within a “Control” vs. “Treated” sample it is only possible to estimate the relative amount of the modified peptide species by a concurrent decrease in the amount of the unmodified peptide. This in essence is a decrease in the proteoform-typic peptide in comparison to the native proteoform-peptides.

**Table 1 proteomes-09-00002-t001:** Direct and indirect methods used to determine NPAA misincorporation.

Analysis Type	Method of Analysis	Ref.
Direct	-MS mass shift analysis	[57]
	-MS/MS peptide spectral analysis	NPD
	-MS-READ	[25]
Indirect	-Hydrolysed amino acid analysis	[41]
	-Radio-labelled amino acid analysis	[40]
	-Amino acid competition studies	[32]
	-tRNA micro-assay	[40]
	-Proteoform isoelectric point analysis	[47]
	-Enzymatic activity assays	[48]
	-Dinitrophenyl assay	[36]

NPD stands for no published data.

**Table 2 proteomes-09-00002-t002:** Amino acids and their biological post-translational modifications.

Site of Modification	Letter Symbol	Modification
Alanine	A	Carbonylation
Arginine	R	Hydroxylation, Phosphorylation, Methylation, ADP-ribosylation, Citrullination, Carbonylation
Asparagine	N	Hydroxylation, Methylation, N-linked glycosylation
Aspartic acid	D	Hydroxylation, Phosphorylation, Methylation
Cysteine	C	Hydroxylation, Phosphorylation, Methylation, Sulfation, Myristoylation, ADP-ribosylation, Nitrosylation
Glutamic acid	E	Phosphorylation, Methylation, ADP-ribosylation
Glutamine	Q	Methylation, Carbonylation
Glycine	G	Myristoylation
Histidine	H	Phosphorylation, Methylation
Isoleucine	I	Methylation, Carbonylation
Leucine	L	Methylation
Lysine	K	Hydroxylation, Phosphorylation, Methylation, Ubiquitination, Myristoylation, ADP-ribosylation, Carbonylation, Malonylation, Succinylation, Glutarylation, Biotinylation
Methionine	M	Hydroxylation
Phenylalanine	F	Hydroxylation
Proline	P	Hydroxylation
Serine	S	Phosphorylation, Methylation, Sulfation, O-linked glycosylation, Carbonylation, Decanoylation
Selenocysteine	U	Hydroxylation
Threonine	T	Phosphorylation, Methylation, Sulfation, O-linked glycosylation, Decanoylation
Tryptophan	W	Glycosylation, Bromination, Quinone
Tyrosine	Y	Hydroxylation, Phosphorylation, Sulfation, O-linked glycosylation, Quinone
Valine	V	Hydroxylation, Carbonylation

The table provides context for how a misincorporation at a particular site could have large biological impacts on cell function [58].

**Table 3 proteomes-09-00002-t003:** Toxic NPAAs, their canonical amino acid homologue, disease, and mass shift for incorporation.

NPAA	Homologue	Mass Shift	Theoretical Immonium Ion (*m*/*z*)	Disease Associated with Toxicity	Source
BMAA	Serine	+13.0316	73.0766	MND [99] AD/PD [100]	Cycad palms and cyanobacteria [101,102]
	Alanine	+29.0266			
L-DOPA *	Tyrosine	+15.9949	152.0712	PD [15]	Velvet bean plant (*Mucuna pruriens*) [103]
	Phenylalanine	+31.9898			
Meta-tyrosine *	Tyrosine	NMS	136.0762	AD/PD [104]	Fescue grass (*Festuca* spp.) [105]
	Phenylalanine	+15.9949			
Ortho-tyrosine *	Tyrosine	NMS	136.0762	AD/PD, marker of Asctheleroisis [104]	Oxidation product of phenylalanine
	Phenylalanine	+15.9949			
Norvaline	Leucine	−14.0157	72.08132	Cytotoxic [61]	Nutritional supplement [61]
	Isoleucine				
	Valine	NMS			
Azetidine 2 carboxylic acid	Proline	−14.0157	56.05002	MS [19]	Lily of the valley (Convallaria majalis);
					Garden beet (Beta vulgaris) [9]
Canavanine	Arginine	+1.9793	131.0933	MS/Systemic lupus erythematosus [106]	Jack bean plant (*Canavalia ensiformis*) [107]
Norleucine	Methionine	−17.9564	86.09697	NNAD	Bacteria
Mimosine	Tyrosine	+16.9902	153.0664	NNAD	Leucaena (*Leucaena* spp.) and some *Mimosa* species [108]

NMS stands for no mass shift. NNAD stands for no named associated disease and * denotes NPAAs that are formed via oxidation of a proteogenic AA. Calculated theoretical immonium ions were assumed to be in positive ionisation mode.

**Table 4 proteomes-09-00002-t004:** Tools for proteomic analysis that can be used in the pursuit of MiP.

Program	GUI	Cost	Open Searches	Accessibility	DIA Searching	Paper
Byonic	Yes	Licensed	Yes	Easy	No	[114]
EncyclopeDIA	Yes	Free	No	Easy	Exclusively	[116]
Fragpipe	Yes	Free to academics	Yes	Easy	Yes	[109]
*Msfragger, Philosopher, PTMShepard*						
Galaxy P	Yes	Free	No	Intermediary	Being implemented	[117]
Mascot	Yes	Licensed	No	Easy	No	[118]
MassIVE	Yes	Free	Yes	Easy	Yes	[119]
Maxquant	Yes	Free	Yes, Dependent peptides	Intermediary	No	[92]
*Andromeda*						
MetaMorpheus	Yes	Free	Yes	Intermediary	No	[91]
OpenMS	Yes	Free	Yes *	Advanced	Yes	[120]
Open-pFind	Yes	Free Licensed	Yes	Easy	No	[110]
Peaks Studio	Yes	Licensed	Yes	Easy	Yes	[121]
Protein Pilot, PeakView	Yes	Licensed	Yes: Protein Pilot	Easy	Yes: PeakView	[122]
Proteome Discoverer	Yes	Licensed	Through Nodes	Easy	Yes	[123]
R workflows *	Yes	Free	Yes	Advanced	Yes	[124]
Skyline	Yes	Free	N/A	Intermediary	No	[125]
Signal quantification (DIA, MRM, PRM)						
SpectroMine	Yes	Licensed	Yes	Easy	No	[126]
*PTM Shepard*						
Spectronaut	Yes	Licensed	No	Easy	Yes	[127]
TagGraph	No	Free	Yes	Advanced	No	[111]
Trans-Proteomic Pipeline	Yes	Free	Yes	Advanced	No	[128]
*PTMProphet*						

GUI: Graphical User Interface. * Consultation of Bioconductor resources is recommended for a grounding in R implemented workflows. Engine/algorithm names are italicised.

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
