# Peer review of "Misincorporation Proteomics Technologies: A Review"

_proteomes, 2021, doi:10.3390/proteomes9010002_

Round 1

Reviewer 1 Report

This review article shows interesting topic and useful information for researchers. The displayed data and figures are clear and self-explanatory and the overall contents are well constructed.

The additional brief explanation for figures could be added in the figure legends. In the present form, there is only simple title.

Author Response

Reviewer 1: This review article shows interesting topic and useful information for researchers. The displayed data and figures are clear and self-explanatory and the overall contents are well constructed. The additional brief explanation for figures could be added in the figure legends. In the present form, there is only simple title.

  • Thank you for your effort in reviewing this manuscript.
  • Figure 1: Now has a sentence describing the process of tRNA AA charging.
  • Figure 2: A more comprehensive description was added.
  • New figure 2: has a descriptive caption.

Reviewer 2 Report

This review is highly interesting and contains information that is important to the field.  It is well-written and provides a useful account on the misincorporation of non-protein amino acids in proteins, as well as the challenges in identifying and locating them. It summarizes current research developments and gaps, and addresses the need for more valuable research outcomes with potential to help prevent or treat human diseases arising from NPAAs disrupting normal functioning of cells. Recommended for publication subject to minor corrections

Check for grammatical and typographical errors throughout the text

Page 4 Line 174 - did you mean  'different sample'

Pages 12-16 Several mistakes exist in the reference list. It should be reformatted thoroughly according to journal instructions. 

Author Response

Reviewer 2:  This review is highly interesting and contains information that is important to the field.  It is well-written and provides a useful account on the misincorporation of non-protein amino acids in proteins, as well as the challenges in identifying and locating them. It summarizes current research developments and gaps, and addresses the need for more valuable research outcomes with potential to help prevent or treat human diseases arising from NPAAs disrupting normal functioning of cells. Recommended for publication subject to minor corrections

Check for grammatical and typographical errors throughout the text

  • These have been corrected.

Page 4 Line 174 - did you mean  'different sample'

  • This has been corrected.

Pages 12-16 Several mistakes exist in the reference list. It should be reformatted thoroughly according to journal instructions

The reference list was created using the MDPI template and is thus in the format required by the journal. Any necessary changes will be performed by the editorial staff during final production.

Reviewer 3 Report

Proteomes Manuscript ID: proteomes-1060097

The review titled: “Mis-Incorporation Proteomics Technologies: A review" is well written and compilation is very informative about the available technologies to study mistranslations and misincorporations proteomics as well as emphasizes on further developments, improvements and refinements of MiP technologies. Authors have explained the impacts of misincorporation of non-protein amino acids on biological functioning of many proteins causing toxicity and disease conditions.

Minor comments:

  1.  Authors needs to include more recent references wherever possible i.eA) The Cyanotoxin and Non-protein Amino Acid β-Methylamino-L-Alanine (L-BMAA) in the Food Chain: Incorporation into Proteins and Its Impact on Human Health. Rachael A Dunlop, Gilles J Guillem; Neurotox Res, 2019 36(3): 602-611.B) Microbial BMAA elicits mitochondrial dysfunction, innate immunity activation, and Alzheimer's disease features in cortical neurons. Silva DF, Candeias E, Esteves AR, Magalhães JD, Ferreira IL, Nunes-Costa D, Rego AC, Empadinhas N, Cardoso SM; J Neuroinflammation. 2020 Nov 5;17(1):332.
  2. Authors should keep same reference pattern throughout the review i.e delete Unimod in Table 1, reference 60, line 202.

Author Response

Reviewer 3: The review titled: “Mis-Incorporation Proteomics Technologies: A review" is well written and compilation is very informative about the available technologies to study mistranslations and misincorporations proteomics as well as emphasizes on further developments, improvements and refinements of MiP technologies. Authors have explained the impacts of misincorporation of non-protein amino acids on biological functioning of many proteins causing toxicity and disease conditions.

Minor comments:

     Authors needs to include more recent references wherever possible i.e A) The Cyanotoxin and Non-protein Amino Acid β-Methylamino-L-Alanine (L-BMAA) in the Food Chain: Incorporation into Proteins and Its Impact on Human Health. Rachael A Dunlop, Gilles J Guillem; Neurotox Res, 2019 36(3): 602-611.B) Microbial BMAA elicits mitochondrial dysfunction, innate immunity activation, and Alzheimer's disease features in cortical neurons. Silva DF, Candeias E, Esteves AR, Magalhães JD, Ferreira IL, Nunes-Costa D, Rego AC, Empadinhas N, Cardoso SM; J Neuroinflammation. 2020 Nov 5;17(1):332.

  • These references have been added to the revised manuscript.

    Authors should keep same reference pattern throughout the review i.e delete Unimod in Table 1, reference 60, line 202.

  • This reference has been deleted

Reviewer 4 Report

This manuscript does an excellent review of the proteomic techniques employed for the top-down proteomic analysis of proteoforms arising from the incorporation of non-protein amino acids (NPAAs). The authors have introduced the mechanism of proteoform formation in the cell and explored the different traditional methods of identifying the misincorporated NPAAs. The review focusses on the use of high-throughput proteome scale methods like mass spectrometry for more efficient and specific identification of the NPAA incorporated peptides

The review starts by describing the process of proteoform generation in the cells and the implications of NPAA in proteinopathies. However, the review fails to mention anything regarding the beneficial mistranslation that leads to proteome diversity and provides adaptive advantages to the cells. Proteome diversity contributing to improved strains with better stress response has been widely reported. The authors address the importance of researchers being able to identify and quantify the MiP by optimizing the sample preparation methods, selecting the mass spectrometry and bioinformatic techniques that are better suited for low abundant proteoforms. But, depending on the mechanism resulting in mistranslation, the effect on the proteome are classified as local mistranslation, regional mistranslation, or global mistranslation. The readers will benefit from a brief paragraph comparing the sensitivity of the detection methods employed to explore protein mistranslation. 

The authors talk about data analysis techniques and recommend the open searching method for MiP. The authors have emphasized the challenge that exists in detecting a small population of misincorporated amino acids in a proteome against the wild type peptides. An unbiased bioinformatics approach is crucial for identifying this minute cohort of mistranslated peptides from a large MS spectra database. They mention a quite comprehensive list of algorithms for performing the open search of the data. They also highlight their point using an example of the methodology developed by Mordet et al. However, a researcher can benefit from the review if he understands the advantages of using these software and how they compare to each other. The big question anybody working with huge data sets is always trying to comprehend how to best extract a signal from their data and how their choice of software can affect the outcome. A table briefly mentioning the features of each of these programs and their approach to separate signals from the background can advise the reader regarding the relative strengths and weaknesses when comparing across multiple techniques. This is a solid review and is well written, providing the readers with information about the technical nuances of applying proteomic techniques for determining the NPAA containing peptides on a proteome scale. The authors discuss a field of proteomics and bioinformatics that will allow us to tackle the technical difficulties arising in the detection of errors in protein synthesis and will help develop therapeutics for aging and human disease in a robust and comprehensive manner. I recommended the manuscript to be accepted after a minor revision.

One typo in the manuscript is

... Line 174-> “different of sample types” to “different sample types”

Author Response

Reviewer 4: This manuscript does an excellent review of the proteomic techniques employed for the top-down proteomic analysis of proteoforms arising from the incorporation of non-protein amino acids (NPAAs). The authors have introduced the mechanism of proteoform formation in the cell and explored the different traditional methods of identifying the misincorporated NPAAs. The review focusses on the use of high-throughput proteome scale methods like mass spectrometry for more efficient and specific identification of the NPAA incorporated peptides

The review starts by describing the process of proteoform generation in the cells and the implications of NPAA in proteinopathies. However, the review fails to mention anything regarding the beneficial mistranslation that leads to proteome diversity and provides adaptive advantages to the cells. Proteome diversity contributing to improved strains with better stress response has been widely reported. The authors address the importance of researchers being able to identify and quantify the MiP by optimizing the sample preparation methods, selecting the mass spectrometry and bioinformatic techniques that are better suited for low abundant proteoforms. But, depending on the mechanism resulting in mistranslation, the effect on the proteome are classified as local mistranslation, regional mistranslation, or global mistranslation. The readers will benefit from a brief paragraph comparing the sensitivity of the detection methods employed to explore protein mistranslation.

  • The authors recognise the positive effects NPAAs can have on cells in terms of increasing cellular survival in organisms such as coli or that the oxidation of amino acids (forming NPAAs) can activate stress responses to enhance eukaryotic survival, and have added a sentence to that effect. However, we think that further elaboration on this point is outside of the scope of this review as our focus is the exploration of disease associated with mistranslations.
  • We have added a paragraph on sensitivity, however it is brief as the sensitivity of techniques is dependent on numerous specific and different factors between the techniques. The proper exploration of this point would require many paragraphs of explanation and some speculation, which would increase the manuscripts length and we don’t believe any further text would enhance the manuscript.

The authors talk about data analysis techniques and recommend the open searching method for MiP. The authors have emphasized the challenge that exists in detecting a small population of misincorporated amino acids in a proteome against the wild type peptides. An unbiased bioinformatics approach is crucial for identifying this minute cohort of mistranslated peptides from a large MS spectra database. They mention a quite comprehensive list of algorithms for performing the open search of the data. They also highlight their point using an example of the methodology developed by Mordet et al. However, a researcher can benefit from the review if he understands the advantages of using these software and how they compare to each other. The big question anybody working with huge data sets is always trying to comprehend how to best extract a signal from their data and how their choice of software can affect the outcome. A table briefly mentioning the features of each of these programs and their approach to separate signals from the background can advise the reader regarding the relative strengths and weaknesses when comparing across multiple techniques. This is a solid review and is well written, providing the readers with information about the technical nuances of applying proteomic techniques for determining the NPAA containing peptides on a proteome scale. The authors discuss a field of proteomics and bioinformatics that will allow us to tackle the technical difficulties arising in the detection of errors in protein synthesis and will help develop therapeutics for aging and human disease in a robust and comprehensive manner. I recommended the manuscript to be accepted after a minor revision.

  • A table has been added listing these programs and software.

One typo in the manuscript is

... Line 174-> “different of sample types” to “different sample types”

  • “of” has been deleted

Reviewer 5 Report

The manuscript described the current technologies of analyzing Mis-incorporation Proteomics (MiP) as well as the need for further development and refinements of relevant approaches.   Overall, it scientifically sounds well.  It is an informative review that deserves publication if the following comments have been addressed.

  1. As a review with only two figures, authors can consider to increase the number of figures to illustrate those assay or biological mechanisms described in the manuscript. For example, it will be more informative to readers if authors can illustrate the cellular prevention, treatment and check mechanisms for NPAA MiP occurrence described in Page2-3.
  2. Similarly, I recommend to tabulate those direct and indirect methods described in the session of “The identification of NPAAs misincorporation”.
  3. The full-term of L-DOPA is missing.
  4. With regards to Table 1, the content was referred to a single reference published in 2004 which might be out of date. I was wondering if the authors can check more and newer individual references for each site of amino acid modification in order to update the content of Modification columns which have been newly discovered after 2004.

Author Response

Reviewer 5: The manuscript described the current technologies of analyzing Mis-incorporation Proteomics (MiP) as well as the need for further development and refinements of relevant approaches.   Overall, it scientifically sounds well.  It is an informative review that deserves publication if the following comments have been addressed.

    As a review with only two figures, authors can consider to increase the number of figures to illustrate those assay or biological mechanisms described in the manuscript. For example, it will be more informative to readers if authors can illustrate the cellular prevention, treatment and check mechanisms for NPAA MiP occurrence described in Page2-3.

  • A new figure has been added demonstrating the fate of mistranslation and the cellular checks involved.

    Similarly, I recommend to tabulate those direct and indirect methods described in the session of “The identification of NPAAs misincorporation”.

  • This table has been added.

   The full-term of L-DOPA is missing.

  • Has been added at first occurrence.

    With regards to Table 1, the content was referred to a single reference published in 2004 which might be out of date. I was wondering if the authors can check more and newer individual references for each site of amino acid modification in order to update the content of Modification columns which have been newly discovered after 2004.

  • This reference was to the original publication of the database itself, which is constantly updating. Each modification added to the database subsequent to the original publication does have a reference supporting the site localisation and can be found on the database. The addition of every paper for the individual sites would add another ~80 papers to this paper, which is unnecessary as researchers can refer to the most up to date information on the website.